# Self and Dignity: The Spirituality of Survival

**Christopher Turner**

Stirling Theological College, University of Divinity, Melbourne 3170, Australia; cturner@stirling.edu.au

**Abstract:** This paper examines the nature of spirit and spirituality as organic response to threat in the context of a global pandemic. Drawing from the fields of neuroscience, philosophy and theology, the author defines spirit as the biological capacity of a living organism to maintain homeostasis in response to changes in its environment. The capacity of individual human organisms to respond to changes that are perceived as threats to homeostasis with passive and active power is posited as a spirituality that is crucial for the survival of the human species. The paper represents a form of secular spirituality that is synonymous with the natural power of organic life.

**Keywords:** spirit; spirituality; power; COVID-19; homeostasis; survival

## 1. Introduction

> . . . *the truth about one's life is outside oneself,*
> *in events, in other people, in things; to*
> *talk about oneself, one must talk*
> *about everything else.*
> *To survive is, after all, perpetually to begin to live again.*
> *I hoped I would still know how.* (De Beauvoir 1957)

The most intense period of 'lockdown' during the COVID-19 response in my home state of Victoria in Australia led to the gathering of my family members in our home so that we could all be together for the duration. In the course of a week our home went from a place of relative space and quietness to a thriving, noisy, socially intense community. My way of survival during this welcome, yet exhausting, situation was to get up in the very early morning and go for a run by myself. It didn't take long before others in the household realised what I was doing. On one of these early mornings someone said, 'can I come?' I felt myself sag on the inside because I just wanted some 'me time'. I said 'yes'. Why did I say that?

It may seem strange that I was desiring personal space during a time of intense limitation of movements, working from home, and general social distancing. My own challenges during the first social response to COVID19 were relatively mild in light of global challenges, particularly in contexts where population density, political agendas and access to medical resources were generating complex hurdles for many communities (Horesh and Brown 2020; Novacek et al. 2020).[1] It was evident to me that whatever I was experiencing would be intensified and contextually shaped by the specific circumstances in which the challenge to stay spiritually healthy was being accepted. Whilst my experience may have shaped my spiritual health, for others the challenges were more immediately threatening to life and relationships. Nevertheless, what I was experiencing was the complexity of maintaining a healthy sense of self in an environment that had changed due to the threat of a virus, and I will argue that this complexity is both contextually unique and common to the spiritual life of all living organisms. How does one assert oneself as

---

[1] Whilst contextual studies into the impact of COVID-19 are relatively few in light of the fact that contextual nuances are still being identified and assessed, some examples that make it clear that unique contextual factors must be taken into account are emerging.

autonomous agent in freely chosen and safe relatedness (attachment) when the usual levels of agency available are drastically altered due to the threat of an infectious virus? The spectrum of states that occur within the dynamics of autonomy and relatedness necessarily shift the basis on which we perceive of ourselves as selves. One of the most disruptive elements of the pandemic is its ability to confuse our sense of self as it is expressed through the interweaving of powers that we think of as autonomy and relatedness. It is not without a sense of irony however, that we note, in fact, that it is under the conditions of response to threat and all other forms of alterity that the sense called 'self' actually emerges as an integrating feeling.

Paul Tillich, developing the line of thought that was affirmed in the theological approach of Friedrich Schleiermacher, illuminated this existential understanding of the self that is established in the power of courage. The elements of autonomy and relatedness are described as the "courage of individualization" and the "courage of participation" in the face of the existential threats of fate and death (Tillich 1962).[2] Schleiermacher went so far as to assert that the feeling of absolute dependence represents the "highest form of self-consciousness" and that this self-consciousness emerges simultaneously with the feeling (Schleiermacher 2011). That assertion demonstrated insight that was consciously reminiscent of Spinoza in that it suggests that selfhood is a unification of pre-conscious physiological processes, processes that both Spinoza and in our own time Damasio, call emotions (Damasio 2004).

It is evident that the requirement for social distancing in the global context of COVID-19 is as complex for our sense of self as is the requirement for social engagement as an unforeseen consequence of 'stay at home' requests that characterised 'lockdown' in Australia and other parts of the world. Whether these requirements generate the challenge of greater autonomy through enforced isolation, or greater relatedness through enforced relational proximity, the question of whether a sense of self that is spiritually robust can emerge in these contexts is pertinent for everyone. What happens to the sense of self when we are required to engage in a level of social distance from family, friends, and community that reduces the presence of the usual social signifiers of selfhood? Conversely, what happens to the sense of self when we are forced into levels of relational proximity that diminish the relational signifiers of selfhood. The threat implied in the COVID-19 pandemic has required both of these dynamics of many in the Australian community and globally, particularly those engaged in the aged care system as residents and families of residents (Berger and Reupert 2020; Luchetti et al. 2020).

The sense of self as the experiential power of autonomy and relatedness that was articulated by Schleiermacher and Tillich is key to the understanding of spirit and spirituality that is argued in this paper. To speak of self in this way is to accept that selfhood is a subjectively experiential reality, though not an observable phenomenon from the so called objective third person perspective (Weger et al. 2016). To accept that selfhood is experienced subjectively and concurrently to accept that, to date, it has proven impossible to observe 'the self' from an objectively scientific perspective, does not necessarily lead us to the conclusion that selfhood is either synonymous with consciousness or a socially constructed illusion (Zahavi 2014). The fact that organisms are multiple preconscious and unconscious physiological processes that engage in constant change in response to both internal and external environmental changes makes it immediately apparent that, if we are to call these processes part of ourselves, selfhood cannot be reduced to consciousness of self or self-perception (Dennett 1996; Graham 1992).

This stance leaves us the necessity of considering that the awareness of self, though experiential and therefore present as internal consciousness, must emerge in the consciousness on the basis of external environmental stimulation (Freeman 2017). It is precisely this conclusion that leads me to speak of the *sense* of self, a designation that allows for

---

[2]  Tillich was developing Schleiermacher's concept of faith as "the feeling of absolute dependence" (relatedness or participation) that necessarily included the feeling of freedom (autonomy or individualization) as a limited expression of power as the affirmation of life in threatening circumstances.

the selfhood of the physiologically pre-conscious organism. The sense of self can be both physiologically pre-conscious and emergent in the sense that consciousness of self can emerge under the influence of relational factors. The latter, though more opaque to the scientific perspective, is posited as the emergent awareness of the "potential self" under the relational condition of "unconditional positive regard" by psychotherapist Carl Rogers and, from the philosophical perspective, the capacity for selves to be "revealed" to the third person observer under the relational condition of 'love without condescension' in the work of Raimond Gaita (Rogers 2004; Gaita 1999). All of these ways of speaking about the self necessarily include what is called the experiential self that emerges in response to external and internal relational and physiological conditions.

It is for this reason that Žižek's reading of Hegel is helpful in understanding both the experiential and the external factors that lead to the sense of self as emergent. For Žižek, the sense of self is a negative sense. Self is sensed unconsciously, pre-consciously, and consciously under the conditions of alterity in relation to the environment of the organism (Žižek 2016). By alterity I mean that kind of 'otherness' that is sensed on the basis of its changing characteristics that in turn generate change in the whole organic self. In other words, I am a self because all of the organic processes that constitute me as an individual organism can sense that I am not you, or them, or it. Most importantly, this basic negative sense is a form of physiological response to the negative environmental signifiers, a response that Damasio argues is basically the function of homeostatic regulation (Damasio 2000). This understanding of self includes the self as sensed in experience both internally (self-consciousness) and externally, in relation to alterity. However, just as Schleiermacher argued that freedom was only possible under the conditions of absolute dependence, so we must accept that internal self-consciousness is completely reliant on the sense of relational and environmental signifiers.

The complex dynamics of isolation and the intensification of social engagement under the conditions of the collective response to COVID-19 constitute a potential crisis of the sense of self that I will argue is a spiritual crisis. The crisis need not be viewed as a necessarily harmful one. The imperceptibility of self in the context of isolation or intense relational engagement need not necessarily signify the dis-integration of the organism as sensed self. Where a sense of self becomes opaque, the processes sustaining a conscious and social will to live are provoked into activity. What I am describing, from the perspective of the consequent social attachments and social and environmental isolation, is a kind of spiritual *provocation*. Spiritual provocation is a state in which the dynamics that promote flourishing life are being threatened and the response of the organism to the perception of threat in the form of organic power is stimulated into pre-conscious and conscious activity. Spiritual vitality occurs when we are able to engage freely in relational attachments (family, community, environment) and individual agency (freedom, space, solitude) in response to changes in our environment. 'Freely' means that we are enabled in our power response to the changes that threaten us. It means that our power to change self and environment in the pursuit of life is accepted, affirmed and validated. The 'will to live' is allowed to flourish as it shapes our activity in the presence of threat (Schweitzer 1936). Nietzsche asserted this with his characteristic criticism of all forms of pity in response to human suffering. For him, it is the wound that provokes the spirit, that shocks the organism into the active and passive power of living (Nietzsche 1989; Young 2010).[3] It is this emergence of organic power as a response to alterity, represented by the "wound" or by Hegel's "negation" as threat, that we will explore as the basis of self as spirit and dignity in this essay (Žižek 2016).

---

3　The use of Nietzsche in the argument for the physiological basis of spirituality is not only warranted but necessary, since to ignore his criticism of religious and philosophical idealism would constitute an unforgiveable blind spot in the development of current understandings of the human spirit. Reservations about Nietzsche that are based on the perception that his philosophy was anti-Semitic should be laid to rest on the basis that it is established beyond doubt that Nietzsche was not anti-Semitic and that in fact he actively despised anti-Semitism in both its political and cultural forms. To omit Nietzsche's perspective because it is anti-religious would be to miss the fact that many of Nietzsche's philosophical conclusions reflect a deep seated concern for genuine spiritual life that was arguable grounded in his early religious devotion.

There are three concepts, therefore, that I will outline in order to give our understanding of spirit and spirituality the depth and breadth required for it to be sufficient for the context of a global pandemic. The first is the concept of threat as the provocateur of spirit, the second is the concept of power as spirit, and the third is the concept of dignity as the agency of the human will to live. When these concepts are seen as the foundations of the human spirit we will be able to conclude that spirituality is the formation of living response to the threat and that the task before us is to determine which elements of our spiritual traditions promote the free expression of human spirit and which elements of our spiritual traditions handicap the spirit or at worst, under the guise of pity or compassion, actually promote decadence.[4]

My personal spiritual response to the state-wide lockdown that required people to stay at home was to change my behaviour in accordance with the science that is driving the Health Department's enactment of policies relating to social distancing in a pandemic. This, as we will see, is a form of power. That was not my only response. I also ran early in the mornings. Running is a spiritual practice for me, and as such, it also is a form of power. It is one of the moments in my day that my brain in collaboration with all my organic systems become vital with the power of movement. I know that not everyone can or wants to run. Living, however, entails the spiritual practice of determining for oneself what kind of response to threat will promote life and the feeling of being alive that we call 'self'.

Having learned the lesson that my run needs to be a solitary spiritual practice, I began to rise in the dark, before the others began to stir, quietly dressed for the run and left the house. The early morning air was crisp, a light breeze blew, and dawn was casting its unique light across the landscape. I ran. By myself. I could feel my heart rate rising, my muscles protesting, my lungs drawing in more and more oxygen. In the quiet dawn, the solitary rhythm of my body allowed my emotions and thoughts to gather in a spacious place in me, uncluttered by social demands. In that space I could feel them, weigh them, and test them without hurry or pressure.

I started to feel like myself again.

## 2. Threat and Spirit

Spirit is the organic process of response to change and threat (Turner 2017). When the existential and therefore experiential understanding of spirit in Schleiermacher and Tillich is brought together with the physiological understanding in Damasio and in trauma studies such as that undertaken by Bessel van der Kolk, it is necessarily located in the preconscious processes of homeostatic regulation in organic life as a response to that which changes or threatens the conditions under which it is possible to be alive (van der Kolk 2014). Understanding this is important if a healthy concept of spirituality in the context of a global health crisis is to be developed. It is a misconception to equate healthy spirituality with peace, safety, happiness, unity, enlightenment, etc. Rather, in line with the understanding of homeostatic regulation as persistently active and reactive in response to changes in the environment that are sensed by the organism (emotion), spirituality is the power response of the organism to a dynamic environment (Marshall et al. 2020).

Another common misconception is that homeostasis is a static state, or at the very least a state of equilibrium that is close to stasis (Frankl 1959; Comas-Díaz et al. 1998).[5] In fact, homeostatic life processes in themselves, and therefore the spirituality of human life, are entirely determined by responsiveness (as change) to changes that are perceived as threats

---

4  Here I intend Nietzsche's meaning of the word decadence, which is, that which ultimately means death for organisms when it is exercised.

5  Conditions that seem unchangeable and that cause considerable human suffering have led to this misconception of homeostasis. Victor Frankl reduced homeostasis to a "tensionless state" that he deemed to be contra the human "struggle for a worthwhile goal" that he placed at the heart of spiritual life. Liberation Psychologist Martín Baró condemned homeostasis for what he saw as the "consecration of the existing order as natural" when the struggle for justice and liberation in El Salvador called for a focus on action towards liberation. Whilst the contextual realities make these views understandable, they miss the foundational characteristic of homeostasis, that is, constant responsive change in order to maintain the conditions under which it is possible to be alive and to flourish.

to homeostasis, as is evidenced in the research into systemic homeostatic fluctuations as integral to human health.

> "*Variability and reactivity are among the most important functional characteristics of physiological regulatory systems. Intrinsic oscillations in system states reflect the activity of homeostatic regulation mechanisms, which allow such systems to flexibly respond to internal and environmental demands and perturbations. Reactivity, that is, short-term changes in physiological states from baseline during changing conditions, reflects the adjustment of bodily functions in order to cope with situational requirements. The ability of organisms to respond flexibly to changing internal and external conditions confers substantial health and survival benefits; abnormalities in variability and reactivity of physiological regulatory systems may weaken the organism, and characterize numerous mental and physical disorders.*" (Duschek et al. 2021)

Contrary to fears that this view is reductionist to determinism, the organism's response to the determining factors of environmental change generate the feeling of freedom through the provocation of organic powers (Nietzsche 1973).[6] Nietzsche's 'will to power' as expression of the 'drives' inherent in human organic existence is an articulation of precisely this. The will to power is not to be understood as a socio-political idea, but as an organic drive to respond to organic threat with organic power; to assert and expend life in the face of threat to life (Nietzsche 1973). Organic life (spirit) drives towards its own survival in the face of threat by the assertion of power. It is this idea that forms the basis of this essay. That which provokes spiritual life is the reality of change and threat. Without change and threat there is no spiritual life. The ideal states of being that are outlined in most religious traditions are entirely imagined states that symbolise the end of threat. Whenever there has been an attempt to realise such states in reality they have been seen for what they are, utopian attempts that are ultimately destructive (Mannheim 1936).[7] For this reason Nietzsche labelled what he perceived to be a religious tendency in modern philosophy (idealism) as decadent (Nietzsche 1954).

To understand threat in this way, as the provocateur of homeostatic power in the organism (spirit), enables us to look at threats to organic life, such as the SARS-CoV-2 virus that results in the COVID-19 disease, as the necessary generators of power, and therefore selfhood, in life rather than forming the view that they form some sort of inevitable existential disaster. The SARS-CoV-2 virus is a threat to the health of human beings that forms part of the environment within which our species evolves, survives, and thrives. It is precisely the nature of the virus as threat that makes it a signifier of the human spirit; the will to live (Schweitzer 1936).[8]

## 3. Power and Spirit

The emergence of the SARS-CoV-2 virus and its interaction with human organisms has disrupted global social, political, and economic systems. When we look at this disruption through the lens of spiritual health, we can see that it has been felt by many as a significant threat to both autonomy and relatedness within relational systems (Thomas 2020). It is understandable that feelings are running high in relation to the multiple and complex threats related to the outbreak of the virus. Whilst many of our feelings are intensely uncomfortable, and may even be life altering when they include the actual events of sickness and death, in fact it is the very nature of the virus as threat to our homeostasis that generates the sense of self as powerful and dignified agent and participant in collective activity in response. In order to understand this, we need to clarify what is occurring in

---

[6] Nietzsche scoffs at the notion of freedom from determining factors in a way that reveals the credulity of the idea that life processes could ever be undetermined. This is not to say that what we call 'freedom' does not exist as part of organic life. It is, however, merely the name we give to the power of the organism in response to determining factors that are working to influence the organism.

[7] Mannheim wrote that, "Only those orientations transcending reality will be referred to by us as utopian which, when they pass over into conduct, tend to shatter, either partially or wholly, the order of things prevailing at that time". The idea of a state in which change and death do not exist is an idea that if it were to pass over into reality it would destroy the biological basis of life itself.

[8] Schweitzer interprets Nietzsche's 'will to power' more precisely in relation to the organic basis of spiritual life.

the human organism when we speak of spirit, and of spirituality, as organic response to change and threat and as the sense of self in the world of relations.

The sense of self common to human organisms is a sense of relatedness to alterity (other) as cause, change, threat, and response (Levinas 1999). Though 'relatedness' sounds like it must be a conscious activity, it is, as we have already seen, predominantly preconscious, or non-conscious, if we think of the activity of the pre-conscious body in the homeostatic processes (Damasio 2000).

The activity of the human organism responding to changes that are sensed as threats to the conditions required in order for life to continue is called spiritual precisely because it is the intrinsic 'will to power', or in the words of Albert Schweitzer, 'will to live' that is irreducible to conscious perception. Organic activity that pre-consciously and consciously pursues the conditions necessary for life is numinous. It is numinous because the changes that take place in us and as a result of our activity when in interaction with the other in our environment, changes that occur as the result of both passive and active power, are irreducible to conscious perception. This occurs not only in us, but also in the other, and the sense of other as powerful limit to our own agency.

It is in this sense that the organic self is said to be numinous. For the sensing subject, 'I' is felt as response to the perception of otherness and as the perception of 'I' as other to the other so to speak. This understanding of the term numinous comes to us again through the tradition of Schleiermacher that was courageously taken up by Rudolph Otto and Paul Tillich. In this tradition there is some basis in Kant's understanding of "noumena" as the form of cognitive understanding that does not rely on the senses in order to understand the object of perception in itself (Kant 1998). Underlying this view is the idea that the senses do not have access to knowledge of an object in itself, only to its appearance as an object in general. This implies an alterity in the object that is not available to the senses. Leaving aside the question of whether there is a form of understanding that can gain access to a perception of an object in itself, it is the element of alterity that is crucial for the understanding of the numinous in this paper.

Rudolph Otto describes the numinous as a "feeling-response" to the perception that the other is "irreducible" to any category that I might use to describe it (Otto 1926). Tillich makes explicit the element of irreducibility expressed in the idea of the numinous by describing it as the " . . . quality of that which concerns man (sic) ultimately (Tillich 1951)". The category of the ultimate in Tillich's thought refers to that which appears as the integrating concern of a person when they perceive the limited, or finite, nature of existence (Tillich 1957). The particular sense of self that each of us is able to articulate is the emergence into consciousness of the particular set of relations (or environmental changes) unique to each person's existence to perception. How exactly does the sense of the self and other as numinous emerge from the organic processes of homeostatic regulation? The answer to this question lies in our perception of the response of the organism as an interweaving of forms of power.

When environmental factors in the human environment, both internal and external, dictate the need for the human organism to change in order to maintain the conditions required for life to continue and to flourish, the multiple processes known as homeostatic regulation occur (Damasio 2004).[9] In this sense, the systems of homeostatic regulation are the biological basis for organic relatedness. The need for change to occur in the organism in the pursuit of life is constant, and as such, the subtle, complex, and multifarious relations between organisms and their environments are constant.

The changes that occur in the individual organism and in its social and biological environment are so complex and extensive that it seems unlikely that they will ever be reduced to a formula, or a scientifically observable or predictable pattern. This is because the internal and environmental changes that occur in the pursuit of life in the organism are

---

[9]　Damasio describes homeostatic processes as changes that take place within the internal environment of the body in response to other changes that occur such as hormonal fluctuations etc. This is what is meant by 'internal as opposed to 'external', which refers to changes that take place outside the physiological boundaries of the individual organism such as temperature etc.

forms of power responding to subtle and multitudinous changes that to a greater of lesser degree are sensed as threats to homeostasis (Whitehead 1978).[10] A 'threat' is any change in the environment, either internal or external, of the organism that requires a corresponding change in the organism in order to maintain homeostasis.

The quality of the numinous is therefore equated with that which is irreducible in the phenomenon of power response in the living organism. The spectrum of powers relating to homeostatic regulation can be broadly categorized in two groups; active powers (relatedness) and passive powers (autonomy). John Locke and Alfred North Whitehead both use these categories in an attempt to understand the complexity of the organism's capacity to respond to change and therefore to threat. Active power, in relation to homeostasis, is the power of the organism to change its external environment in the pursuit of life. These forms of power can be understood as consciously relational. Passive power is the organism's power to change its internal environment, the organism itself, in response to changes occurring both externally and internally that threaten the conditions under which it is possible to stay alive (May 1972). The larger proportion of passive power occurs pre-consciously in the realm of biological and neurological processes, though not exclusively.[11]

The living organism cannot be reduced to any particular form of power, whether active or passive. Life is an irreducible correlation of the full interweaving of powers relating to homeostatic regulation. The irreducibility of interwoven powers in the living organism is what is understood as spirit in pursuit of the conditions necessary for life. Spirit emerges into human consciousness through emotion and feeling. Emotion, which we might also venture to call the sensation of change, and feeling, the conscious awareness of emotion, signal the need for responsive change in the pursuit of the conditions necessary for life (Damasio 2004). Responsive change is determined by the activation of whichever combination of powers will most effectively achieve the conditions necessary for life to flourish. Many factors determine how effective or otherwise the activation of powers is in achieving the goal of flourishing life.

If spirit is the homeostatic response to threat then spirituality is the activation of the integration of the combination of feelings known variously as autonomy and relatedness, agency and resignation, subject (self) and inter-subjective relations. Each of these feeling states is the conscious manifestation of an irreducible interweaving of active and passive powers exercising their influence over external and internal environments in the pursuit of the conditions necessary for the flourishing of life in conditions that represent constant change and therefore threat to life.

The powers at work uniquely in each individual organism generates a sensation of the alterity between organisms in the context of social interaction. When two organisms encounter each other there is always an inter-subjective awareness between them that the other represents change and potential threat that requires a homeostatic response. To think about inter-subjectivity in this way maintains the irreducible otherness that Zahavi argues is crucial if we are to avoid the error of dissolving inter-subjective relations into a false unity (Zahavi 2014). The irreducibility of the individual organism as other, has the power to alter the perception and generate change in the organisms that fall within range of its activation of powers. Here, again the sense or *perception* of the numinous is present in true inter-subjectivity (Tillich 1951).[12]

To summarize, the numinous is the perception, both pre-conscious and conscious, of power in alterity as catalyst that generates change (emotion, feeling, metabolic response,

---

[10]    In referring to the complexity of causes perceived as threat the intersubjective awareness labelled by Whitehead as 'prehension' is drawn upon. Prehension, in Whitehead's organic philosophy, is the category of all elements of 'relatedness' between actual entities. Causality is one of those elements of relatedness. Organic process is built on the imperceptible complexity of prehensions that are often only detected as negative signifiers.

[11]    Alfred North Whitehead, *Process and Reality* (New York: The Free Press, 1978), pp. 58–59. Whitehead uses Locke's ideas of power to describe his philosophy of organism in which ever changing relations are manifest as the exercise of power in interaction and the ever consequent changes. The correlation of passive power with the capacity to change oneself (autonomy) is demonstrated in Locke's analogy of the power of gold to become a liquid under the right temperature conditions (quoted in Whitehead).

[12]    Tillich references Rudolph Otto in relating the numinous to the experience of the holy in people and objects. In doing so he refers, not to moral goodness but, to the experience of the quality of alterity conveyed through the irreducibility of the person or object that is experienced.



neurological response, and conscious reaction) in the one in whom awareness of that power resides (the subject). It has been argued that this awareness is the generator of the integrating sense of selfhood, for the 'other' is any catalyst for change in the environment to which 'I' as organism, respond with change through passive and active power (Damasio 2000). In this sense the numinous is that which animates (provokes a reaction) the individual organism and the intersubjective encounter. The numinous is, for this reason, often described as spiritual. The spiritual life as that which is numinous can therefore also be understood as the perception of alterity as limit. It is limit as change and or threat that gives rise to what we call the sense of self. In fact, the consciousness of self is an integrating construct for the sensible perception of the limiting factor of the numinous in the other (Damasio 2000).

## 4. Dignity

Dignity is the free capacity of the organism to exercise its unique interweaving of passive and active powers in response to that which threatens its ability to stay alive. Dignity is the free capacity to be both autonomous agent and related agent. Crucially, dignity is defined by the capacity and the freedom as an autonomous agent to form safe attachments (relatedness). This definition of dignity arises directly from the understanding of spirit as the homeostatic extension of powers in the pursuit of the conditions necessary for a flourishing life that every organism engages in. As such, dignity and the sense of self are correlated and exist as ideas co-informing each other in the integration of the organism's response to alterity in the world.

Dignity is a concept that is elusive and difficult to define, as is evidenced by the considerable diversity of definitions and disagreement about its veracity as a concept at all (Franeta 2019). Usually tied to the idea of inalienable human rights, dignity is seen variously as an aspect of morality or autonomy in social political states. Kristi Giselsson posits that dignity is " . . . inherent and equal worth . . . linked to the concept of justice: an account that affirms that all human beings can be viewed as ends in themselves rather than as instrumental means to other's ends" (Giselsson 2018). Such ideas of dignity are extremely worthwhile in revealing ethical intent in human relations. Their fragility as concepts leaves us, however, with the same suspicion articulated by Gaita who argued that,

> "*Natural though it is to speak this way, and although it has an honoured place in our tradition, it is, I believe, a sign of our conceptual desperation and also of our deep desire to ground in the very nature of things the requirement that we accord each human being unconditional respect. To talk of inalienable dignity is rather like talking of the inalienable right to esteem. Both are alienable; esteem for obvious reasons, and dignity because it is essentially tied to appearance.*" (Gaita 1999)

The attempt to ground dignity in the human spirit as the homeostatic 'will to live' *is* an attempt to ground it in the very nature of things. It is, nevertheless, also to recognise that it is possible to be alienated from one's own capacity to pursue life as the freedom of homeostatic autonomy and safe relatedness. When we witness the alienation of another person from their dignity it takes a unique kind of love for the other as one's equal in the quest for life to reveal their dignity when brutality or force of circumstances renders them powerless to appear dignified in the presence of others. Schweitzer believed that the most atavistic and physiological forms of spirit manifest as the "will to live" that produces a universal ethical sense that he called "reverence for life" (Schweitzer 1923). His argument grounded the reverence for life in the homeostatic inter-dependence of every individual organism with other living organisms, a kind of social homeostasis. With Schweitzer we can argue that if dignity is a natural element of spirit as the homeostatic will to live, then the struggle for life in the face of threat to life is always and in every form, dignified.

The idea of dignity in relation to death necessarily fits within this definition, emphasising the social perspective of organic relatedness. A virus does not only threaten other organic life through the challenge to its powers of homeostatic response, a virus sometimes kills other organisms. The social perspective of homeostasis reveals that in fact, the death

of an individual organism is a form of power in the homeostasis of the species and the environments of the species. This is a challenging idea when those we love are at risk of death or in fact have died. It is an idea that establishes the dignity of death as a form of social homeostasis however. As such it is an idea with significant meaning for all human beings, and particularly those who do not accept the religious ideas of life after death.

COVID-19 has presented human organisms with significant challenges to elements of dignity. Human responses to the pandemic range from the reasoned and cooperative, to the withdrawal into conspiracy theories that may feel like autonomy in the context of collective social action. Societies that have effectively responded to COVID19 have accepted the demand to voluntarily distance themselves from one another (Blau and Tonkin 2021). This response is simultaneously an expression of the passive power of autonomy and the active power of asserting the self in social cooperation. That which is a demand for individuals to exercise autonomy by distancing themselves is simultaneously a demand for relatedness in the exercise of the social responsibility to ensure the safety of others.

One of the key challenges for societies that are contemplating how to organise considerable social action in response to the threat of COVID19 is the challenge of garnering social response through the genuine agency (the genuinely free ability of each person to respond with passive and active power to the threat of the virus) of each member of the society. Societies that feel empowered to respond with the interweaving of passive and active powers available to them will feel that they are able to respond in a dignified manner, and thus retain a strong sense of self. Individuals who respond with dignified power to the threat will be responding on the basis of an active spiritual life and will therefore feel both autonomous and relational power as they participate in the collective societal response.

## 5. A Myth of Power and Dignity

Thinking of the power available to each living organism and its capacity to change itself and its environment in the pursuit of the conditions for thriving life, one of the myths that fills the story telling of the Christian tradition with insight and meaning comes to mind. The myth tells of one of those strange passing moments in the travels of Jesus, the Nazarene, a shadowy figure who is spoken of in so many strange ways throughout the traditions of Christianity. The gospel of Mark (5:23–34NRSV) tells the story in its fullest version, detailing the exchange of powers between an unnamed woman seeking healing and the Nazarene (Coogan 2001). The story goes that a woman who has been menstruating for twelve years, a condition that implies her religious and therefore social exclusion, decides to break the purity laws of her religion in pursuit of healing. A man with the reputation of a healer is passing and she enters the throng in the hope of touching his garment and thereby taking her healing from him by stealth. It is worth noting the woman's exercising of her own agency in the face of religious prohibition. Applying the ideas about the nature of spirit and spirituality that we have explored so far to this story, we can see clearly that the unnamed woman's "will to live" is given expression in her "will to power" through the assertion of an interweaving of autonomy and relatedness in the face of exclusion. Her spirit overcomes her religion.

The response of the Nazarene is also worth noting when we seek to understand spirituality in the context of threat. The touch of the unnamed woman meant that his religious purity was compromised according to the purity codes (Myers 1988). It was not in Jesus' interest, if you take the religious view of his context, to draw attention to the exchange of power that had taken place. Nevertheless, this is precisely what he does. Additionally, when he sees the woman and hears her story he does not denounce her for breaking the religious law but pronounces her actions to be the actions of faith. In the eyes of the Nazarene, the woman's spirit has overcome her religion and revealed genuine faith; the will to live.

The elements of this myth that have particular importance for the understanding of spirituality are those relating to the phenomenon of power in both the woman and Jesus, and the allocation of the phenomenon of power to the category of 'faith'.

Power is explicitly described as a physical event in the bodies of both Jesus and the woman and perceived as such by them both. Yet, power was phenomenologically evident in the woman before she touched Jesus' clothes. The woman's agency in her actions, contra her religious situation, the interweaving of active and passive powers that place her in the path of healing, and her re-action to the decadence of the religious laws that were disempowering her, are all examples of the human spirit as power response to change/threat. She overcomes the shame inherent in her social exclusion with agency so shocking to those within the shame and honour system of her situation that it is tantamount to a scandal (Myers 1988).

Scandal it may have been, one among many in the life of the Nazarene, yet as myth its power resides in how it reveals the human spirit as irreducible to religion. In the context of the global COVID19 pandemic, the story of the unnamed woman reads as an invitation to face the potential for diminishment in the disease and respond to it with spirit; with power. Though the woman remains without a personal name in the story, her power in the face of threat *is* named. The unnamed woman is first named 'daughter', inverting her social status from excluded to related, and is then named as 'faith' for us. Through the eyes and pronouncement of the Nazarene, we now see the woman not as an unnamed outcast but as an exemplar of vital spiritual life. In this story, the way the crowd views the woman is changed by the way the Nazarene sees her. His view of her reveals her humanity to those whose response was otherwise to exclude her (Gaita 1999). It is a moment of spiritual transformation for everyone present and indeed for us as we read the story.

In this we can see that when the 'other' is encountered in inter-subjectivity, and change and response occurs, the selfhood of the woman emerges as a unifying sense that establishes her dignity in the world. We may well ask whether the capacity to resist her religion could simply be an example to those who would resist the social demand to participate in the social distancing required to mitigate the threat of the virus. The answer would be yes it could. The woman literally ignored the social distancing laws of religion that related to her condition. She did so with the homeostatic power available to her; her 'will to live'. In light of this, individuals must be positively engaged in the discernment process that relates to the dignified activity of both autonomy *and* relatedness in the pursuit of health. In the case of the menstruating woman the only danger to the crowd was one of imaginary impurity. In the case of COVID-19 the danger to the crowd is a virus that can kill. The appeal to the human spirit is to pursue life for self and other, to "will to live amongst other wills to live" (Schweitzer 1936).

Healing is not always guaranteed of course. There always comes that moment in the experience of the organism when it encounters changes to the environment that make it impossible to sustain the conditions required to go on living. However, is dignity lost when this occurs? Is the human spirit extinguished? As Victor Frankl has observed on the basis of his extraordinary experience of threat that feels insurmountable,

> "*When we are no longer able to change a situation—just think of an incurable disease such as inoperable cancer—we are challenged to change ourselves.*" (Frankl 1959)

Spirit and dignity are present even when the threat we face will ultimately take life from us. In these types of circumstances, the passive power of the human will to live is most evident. Like gold that has the power to become liquid when the temperature in its immediate environment is high enough, the human spirit has the power to see the meaning of its life in the broader context of the human story (Whitehead 1978). In this we can see again the nature of power as the interweaving of passive and active elements. For in death, a passive power, active power is also present as relatedness to other lives. The death of each living organism is necessary for the sustaining of organic life itself. The death of each individual organism will ultimately create space for the lives of other organisms. Albert Schweitzer understood this when he deduced the moral element of biological life as " ... the will to live amongst other wills to live ... ", thereby acknowledging that passive and active powers within the organism find their limit in encountering alterity

(Schweitzer 1936). As such, spirit does not exclude death from itself but is in fact most poignantly evident in death as the instinct for the survival of the species.

## 6. The Spirituality of Survival

I have outlined three foundational concepts of spirit and spirituality that help us to think of the human spirit in ways that I argue are sufficient for a healthy spiritual response to a global pandemic, let alone the countless other threats to life that we live with every day. The perception of threat to the conditions required for life to continue provokes an interweaving of active and passive power responses that are the catalysts for the sense of self as dignified agent. This process of homeostatic regulation is the human spirit adapting in the act survival; "the will to live amongst other wills to live".

The spirituality of survival will empower individuals and societies to respond to the threat of COVID19 without losing their dignity. The existential threat implicit in a potentially deadly and contagious virus is met by organisms, both individual and collective, through the exercise of powerful agency in change as a spiritual response. In fact, the self and the dignity of the self are born as the organism changes through the activation of passive and active powers in response to the existential threat.

Those who seek spiritual wisdom from religion in times like this will find it in the aspects of those traditions that enable and empower individuals and communities in the process of powerful change. Where religions emphasise the agency of individuals and communities they have something to speak into this situation. The story of the woman named 'faith' on the basis of her powerful response to threat is a case in point. Where religions ask the individual to abdicate their own subjectivity, and therefore their own agency and power in favour of an idea that does not exist in anything other than the imagination, they can offer only decadence; the opposite of spirit.

A case in point from my own religious tradition is the idea that as a Christian I have 'died' and now live vicariously through the messianic other, who is the only true subject. This idea, most starkly stated in the Pauline letter to the Galatians (2:20), is the kind of idea that Nietzsche described as decadent in that it strips the subject of its own subjectivity, a state of being that Simone De Beauvoir described in her examination of womanhood (De Beauvoir 2010). The self is lost in this scenario and the power of change in response to threat is abdicated to an abstract principle (Christ). It is up to each person formed by the Christian religion to discern between the tradition of the woman and the Nazarene as formative for an understanding of spiritual life, and the idea of the abdication of subjectivity in the Pauline approach of Galatians as formative for an understanding of spiritual life. For the individual human organism to stand in this place of discernment in pursuit of life is what faith actually means.

As the COVID19 pandemic continues, the human spirit will continue to be provoked to response. The hope of humanity lies in its capacity to choose to go on living. The choice is one of discernment in claiming the practices and traditions that enable each one of us to freely exercise our bio-spiritual powers of homeostatic change in the pursuit of life.

**Funding:** This research received no external funding.

**Conflicts of Interest:** The author declares no conflict of interest.

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
