# Peer review of "Self and Dignity: The Spirituality of Survival"

_religions, doi:10.3390/rel12040233_

Round 1
Reviewer 1 Report
This is an intriguing project. The essay is well written and manifests creative, thoughtful reflection. In my view, however, the essay in its current form fails, mainly because the author is more eager to make creative assertions than to build a case that builds on other current literature.
The author uses terms such as “spirit,” “spirituality,” “self,” “numinous,” “faith,” and “dignity” in an idiosyncratic manner. He/she needs to put more effort into acknowledging more mainstream definitions, arguing for why they are inadequate, and then justifying the chosen definitions. There is a vast literature on these topics (not only in religious studies but in philosophy, psychology, and neurobiology) and while no one could master it all, this essay insufficiently references other writings that present similar or related theories.
The author relies heavily on Nietzsche, a choice which also needs to be more thoroughly justified in view of serious questions about perceived negative implications of Nietzsche’s theories. Again, the literature should be engaged and referenced.
While using the pandemic as an example of “threat” that provokes creative response is timely, this way the threat is presented remains highly elitist. A more serious engagement with how the threat is confronted by those not privileged to have a country house and time to take long runs is needed for the essay to make a genuine contribution to humanity’s response to the pandemic. The proposed conclusions in this regard need better development.
A small point: I have never seen “complect” used as a noun, nor does that use seem to be referenced in standard dictionaries. Would “complexus” or “interweaving” be a better term, rather than this idiosyncratic term?
Overall, I confess that I personally disagree with many of the author’s points, but at the same time I affirm the potential validity of this as a theoretical stance that could be justified on the basis of recent literature in various fields. I suggest a rewriting with greater depth and references.
Author Response
My gratitude goes to the reviewer for their time and expertise in offering a critical assessment of the paper. The reviewer’s comments are extremely helpful and pertinent and I anticipate that addressing the points raised will strengthen the paper considerably. Please find the changes made in response to each point raised by the reviewer tabled below.
Reviewer’s Comment |
Author’s Response/Changes |
1. ...the essay in its current form fails, mainly because the author is more eager to make creative assertions than to build a case that builds on other current literature.
|
The reviewer’s comment is pertinent and I have sought to respond to it by more adequately building an argument for the ideas put forward in the paper with reference to other literature. See primarily the changes made in relation to comment two. |
2. The author uses terms such as “spirit,” “spirituality,” “self,” “numinous,” “faith,” and “dignity” in an idiosyncratic manner. He/she needs to put more effort into acknowledging more mainstream definitions, arguing for why they are inadequate, and then justifying the chosen definitions. There is a vast literature on these topics (not only in religious studies but in philosophy, psychology, and neurobiology) and while no one could master it all, this essay insufficiently references other writings that present similar or related theories.
|
The reviewer is correct to highlight these terms as central and to the argument and idiosyncratic in nature by virtue of the particular definitions posited.
Please refer to changes in relation to,
- the theory of self on pages 2-4. - the theory of spirit on page 5. - the theory of the numinous on page 8. - the theory of dignity on pages 10,11.
It has not been possible to systematically assess all other theories within the scope of the paper, however I have sought to demonstrate how each definition and theoretical position has been constructed from the literature and the tradition. |
3. The author relies heavily on Nietzsche, a choice which also needs to be more thoroughly justified in view of serious questions about perceived negative implications of Nietzsche’s theories. Again, the literature should be engaged and referenced.
|
See footnote 15. I have used the outstanding work of Julian Young to resolve concerns about Nietzsche’s perceived anti-Semitism and the use of a philosopher so overtly anti-religious. Furthermore, since the rewrite I believe the paper is less reliant on the work of Nietzsche. |
4. While using the pandemic as an example of “threat” that provokes creative response is timely, this way the threat is presented remains highly elitist. A more serious engagement with how the threat is confronted by those not privileged to have a country house and time to take long runs is needed for the essay to make a genuine contribution to humanity’s response to the pandemic. The proposed conclusions in this regard need better development.
|
Thank you for your perspective on this. I understand that my socio economic position is, relative to many positions globally, privileged. I have altered the contextual details in an attempt to demonstrate that my context is merely an example that, though adequate for me, clearly needs contextual consideration in its translation to other contexts. See changes on page 1. See Footnote 2. |
5. A small point: I have never seen “complect” used as a noun, nor does that use seem to be referenced in standard dictionaries. Would “complexus” or “interweaving” be a better term, rather than this idiosyncratic term?
|
In the interests of clarity I have accepted your suggestion to use “interweaving” as a substitute for the somewhat obsolete “complect”. I thank you for your suggestion. |
Reviewer 2 Report
I found the article well-written and compelling. As a scholar of Christian spirituality who often refers to human spirituality in general it gave me some very though-provoking insights that i will need to further develop for my own work.
I would have liked to see references to works and scholars beyond the euro-centric usual suspects. Perhaps looking into the work of Latinamerican, African and Asian scholars. Still your work is excellent.
Author Response
My gratitude goes to the reviewer for their time and expertise in offering a critical assessment of the paper. The reviewer’s comments are extremely helpful and pertinent and I anticipate that addressing the points raised will strengthen the paper considerably. Please find the changes made in response to each point raised by the reviewer tabled below.
Reviewer’s Comment |
Author’s Response/Changes |
I found the article well-written and compelling. As a scholar of Christian spirituality who often refers to human spirituality in general it gave me some very though-provoking insights that i will need to further develop for my own work.
|
Thank you for these comments. |
I would have liked to see references to works and scholars beyond the euro-centric usual suspects. Perhaps looking into the work of Latinamerican, African and Asian scholars. Still your work is excellent.
|
I think this is a pertinent observation and the paper would certainly be stronger for a wider contextual approach. Though it does not really adequately address your concern I have added reference to Salvadorian liberation psychologist Martín Baró in the discussion about homeostasis as the activity of change and response in pursuit of the conditions. See the discussion on page 6 and footnote 19. |
Reviewer 3 Report
I think the essay starts from an original perspective ("secular spirituality that is synonymous with the natural power of organic life"), and a very relevant application to the pandemic situation we are experiencing.
However, I believe that there is an important problem of justification and integration with other perspectives.
The “complect of powers that we think of as auntonomy and relatedness” is an element that needs more justification.
There are other powers or “highest-order directions in Cooper’s terms that can be contemplated: the area of ​​competence (Ryan & Deci, 2000) also named as self-esteem (Epstein, 1994), achievement (McClelland & Burnham, 2008), and the area of ​​security (Griffin & Tyrrell, 2013) or safety (Maslow, 1987).
The association between active powers with relatedness and passive power with autonomy also need further clarification and justification.
The possibility of the existence of social homeostasis also need further justification (no reference is given)
The concept of homeostasis is presented and the reference to Damasio is appreciated, but there are gaps in justification. Especially in the paragraph: "When environmental factors in the human environment, both internal and external, dictate the need ..." More references are needed to justify these perspectives.
The concept of self is presented integrated in Ĺ˝iĹľek approach, but there are missing very important perspectives (Fritz Perls, Carl Rogers, Karen Horney …)
There is an implcit polarity (religious laws vs, genuine faith) that needs further clarification. The collective appears at times as the place where the individual agent participates, at other times as the bearer of an illusion (religious laws).
Minor issues
Spirit as the "homeostatic power in the organism" but on other occasions as "organic life (spirit)". It could be a typo.
Information is missing (in page 9, “Power (“), there is a type, a bracket needs and end braket.)
Cooper, M. (2019). Integrating counselling & psychotherapy: Directionality, synergy and social change. Sage.
Author Response
My gratitude goes to the reviewer for their time and expertise in offering a critical assessment of the paper. The reviewer’s comments are extremely helpful and pertinent and I anticipate that addressing the points raised will strengthen the paper considerably. Please find the changes made in response to each point raised by the reviewer tabled below.
Reviewer’s Comment |
Author’s Response/Changes |
I think the essay starts from an original perspective ("secular spirituality that is synonymous with the natural power of organic life"), and a very relevant application to the pandemic situation we are experiencing.
|
Thank you for these encouraging comments. |
However, I believe that there is an important problem of justification and integration with other perspectives. The “complect of powers that we think of as auntonomy and relatedness” is an element that needs more justification.
|
Excellent and pertinent observations. I have made considerable changes in the discussions around selfhood, spirituality, and dignity in ways that draw in a greater range of perspectives in an attempt to demonstrate where the argument of the paper fits within the broader and the historical discussion. |
There are other powers or “highest-order directions in Cooper’s terms that can be contemplated: the area of ​​competence (Ryan & Deci, 2000) also named as self-esteem (Epstein, 1994), achievement (McClelland & Burnham, 2008), and the area of ​​security (Griffin & Tyrrell, 2013) or safety (Maslow, 1987).
|
Whilst I have not included these specifically I hope that the more detailed outline of selfhood and its relation to homeostatic change will suffice to both justify the argument and demonstrate that the powers mentioned her fall within the broader category of passive and active powers posited by Whitehead and Locke. |
The association between active powers with relatedness and passive power with autonomy also need further clarification and justification.
|
I have sought to address this concern in footnotes 33 and 34. The clarification is specifically reliant on John Locke’s analogy. |
The possibility of the existence of social homeostasis also need further justification (no reference is given)
|
I have sought to strengthen this in the changes made to the development of the ideas about dignity. |
The concept of homeostasis is presented and the reference to Damasio is appreciated, but there are gaps in justification. Especially in the paragraph: "When environmental factors in the human environment, both internal and external, dictate the need ..." More references are needed to justify these perspectives.
|
You are right to identify a gap in relation to this designation. Please see the added footnote number 32. I have again used Damasio. |
The concept of self is presented integrated in Ĺ˝iĹľek approach, but there are missing very important perspectives (Fritz Perls, Carl Rogers, Karen Horney …)
|
I have made considerable changes to the development of the argument about selfhood. Please see changes on pages 2-4. |
There is an implcit polarity (religious laws vs, genuine faith) that needs further clarification. The collective appears at times as the place where the individual agent participates, at other times as the bearer of an illusion (religious laws).
|
Thank you for your helpful observation on this point. I have sought to address this polarity by including further analysis of the nature of dignity, the argument for its basis in nature, and as such, its capacity to shape human spirituality in an ethical sense. The inference here is that spirit as organic life can and does inform more broadly social conventions of morality. I have removed reference to “illusion” as I think that your observation is right that it creates an unhelpful polarity. |
Minor issues Spirit as the "homeostatic power in the organism" but on other occasions as "organic life (spirit)". It could be a typo. Information is missing (in page 9, “Power (“), there is a type, a bracket needs and end braket.)
|
This has been corrected.
This has been corrected. I had placed (dunamis) in the Greek in brackets but this has been deleted as unnecessary. |
Round 2
Reviewer 1 Report
The revision responds to my concerns. My only suggestions at this point are:
- Break some of the extraordinarily long paragraphs into smaller sections. This helps the reader digest difficult material.
- In some places revisions resulted in poor sentence structure or repetition. It needs one more editing for this.
- The term "complect" reappears in the last few pages. If it is going to be used, it will have to be defined. Otherwise, change it.
Author Response
The author thanks the reviewer for their time and expertise in attending to what is required for strengthening the paper.
Your work and expertise is again very much appreciated in relation to this paper. It is always helpful to receive honest and critically helpful feedback that serves to strengthen the paper. Please see changes made in the table below.
Reviewer’s Comment |
Author’s Response |
Break some of the extraordinarily long paragraphs into smaller sections. This helps the reader digest difficult material.
|
This has been done throughout the document as evidenced in this last edit. |
In some places revisions resulted in poor sentence structure or repetition. It needs one more editing for this.
|
Changed to - It is in this sense that the organic self is said to be numinous. For the sensing subject, ‘I’ is felt as response to the perception of otherness and as the perception of ‘I’ as other to the other so to speak. Page 7.
Changed to - Here again the sense or perception of the numinous is present in true inter-subjectivity (Tillich 1951). Page 9.
Deleted - As Rudolph Otto observed, even inanimate objects can enter the perception as numinous because of their power to generate change, emotion, and feeling in the organism that perceives them. The numinous is the sense of change in the organism as subject when it perceives and experiences the irreducible activation of powers in the other to which it responds. In this sense, as Otto argues, the numinous is both a quality of the object of perception and a “state of mind” that arises in the subject who is within the sphere of influence of the object’s powers (Otto 1926). Page 9.
|
The term "complect" reappears in the last few pages. If it is going to be used, it will have to be defined. Otherwise, change it.
|
All instances of the term complect have now been replaced with the term ‘interweaving’. |